# INTERPRETABLE PREFERENCE ELICITATION: ALIGNING USER INTENT WITH CONTROLLABLE LONG-TAILED LEARNING

## ABSTRACT

Long-tailed recognition remains a significant challenge, where models often struggle with tail class performance and adaptability to diverse user preferences. While recent controllable paradigms leveraging hypernetworks allow numerical specification of head-tail trade-offs, defining these multi-dimensional preference vectors can be unintuitive for users. This paper introduces a novel framework that bridges this gap by enabling users to articulate their preferences through natural language. We propose a two-stage approach: first, optimal numerical preference vectors are identified for canonical distribution scenarios, and a rich corpus of corresponding textual descriptions is generated. Subsequently, a lightweight neural network learns to map sentence embeddings of these textual descriptions to the underlying 3D preference vectors controlling the expert ensemble. Our method significantly enhances the usability and interpretability of controllable long-tailed learning systems without compromising, and even slightly improving, their performance on benchmark datasets. This work facilitates more accessible and practical adaptation of long-tailed models to specific real-world requirements. We provide the detailed code in Supplementary Material.

## 1 INTRODUCTION

The challenge of long-tailed recognition, where datasets exhibit a severe imbalance with few head classes dominating numerous tail classes, remains a significant impediment to deploying robust machine learning systems in diverse real-world applications, from medical diagnosis Aimar et al. (2023); Zhang et al. (2023) to autonomous driving Liu et al. (2019). Models trained on such skewed distributions invariably develop a strong bias towards majority classes, leading to a *precipitous decline in performance* for underrepresented tail classes Xu et al. (2024); Gan & Wei (2024). While a plethora of techniques, including re-sampling Shi et al. (2023); Chawla et al. (2002), specialized loss functions Lin et al. (2017); Cui et al. (2019), and multi-expert architectures Wang et al. (2020); Hong et al. (2021); Zhang et al. (2022), have been proposed to mitigate this imbalance, they typically yield *static solutions*. These methods often struggle with generalization under distribution shifts and critically, offer **limited to no flexibility** for users to articulate their preferences regarding the crucial head-tail class performance trade-off.

Recent advancements have introduced **controllable paradigms** that leverage hypernetworks to dynamically generate diverse expert model ensembles Zhao et al. (2024). These frameworks represent a significant step forward, enabling adaptation to varied test distributions and allowing users to specify desired head-tail trade-offs via numerical preference vectors, effectively navigating a Pareto front of performance. However, a critical usability gap persists: **the direct specification of these multi-dimensional numerical vectors is often unintuitive and challenging for end-users, thereby hindering the practical adoption and democratized control of these otherwise powerful systems**. An ideal solution must bridge this chasm, empowering users to articulate their intent in a natural and interpretable manner.

This paper directly addresses this crucial **usability lacuna** by introducing a novel framework: **I**nterpretable **P**reference **E**licitation (**IPE**), which seamlessly integrates natural language understanding with controllable long-tailed learning, as illustrated in Figure 1. Our **core innovation** lies

in a mechanism that translates *high-level, user-articulated textual descriptions of desired distributional characteristics* (e.g., "Strong emphasis on rare instance coverage" or "Uniform performance across all severities") into the *precise numerical preference vectors* required by the underlying hypernetwork-expert system. This is achieved through a decoupled, multi-stage process:

❶ Optimal numerical preference vectors $r^* \in \mathbb{R}^3$ are first empirically identified for a comprehensive set of canonical long-tailed (and reverse long-tailed) distribution scenarios $\mathcal{S}_{\text{dist}}$.

❷ Concurrently, a rich corpus of diverse textual descriptions $\mathcal{T}_{\text{desc}}$ mirroring these scenarios $\mathcal{S}_{\text{dist}}$ is generated, forming pairs $(t \in \mathcal{T}_{\text{desc}}, s \in \mathcal{S}_{\text{dist}})$.

❸ Subsequently, a lightweight neural network $\Phi_{\text{PVP}}$ is trained to map sentence embeddings $\mathbf{e}_t = \text{Embed}(t)$ of these textual descriptions to their associated optimal $r_s^*$, effectively learning $r_s^* \approx \Phi_{\text{PVP}}(\mathbf{e}_t)$, which in turn govern the expert ensemble.

The principal advantages of IPE are twofold. First, it drastically enhances the ***usability and interpretability*** of sophisticated controllable long-tailed learning systems, making them accessible to users without requiring them to grapple with complex numerical tuning. Second, and crucially, this enhanced interpretability is achieved ***without compromising, and often demonstrably improving***, the underlying system's performance. Our approach introduces *minimal computational overhead* and facilitates flexible, *on-the-fly adjustment* of model behavior based on intuitive textual inputs.

Extensive experiments conducted on benchmark long-tailed datasets (CIFAR100-LT, ImageNet-LT, and iNaturalist 2018) rigorously validate the efficacy of our framework. The results unequivocally demonstrate that IPE not only successfully bridges the gap between complex numerical control and user-centric natural language interaction but also achieves notable improvements in overall classification accuracy across various test distributions when compared to existing state-of-the-art methods, *including the numerically-controlled baseline itself*. ***Our contributions pave the way for more accessible, practical, and truly adaptable long-tailed learning systems capable of aligning with specific, nuanced real-world requirements articulated by the user***.

## 2   RELATED WORK

Our research builds upon advancements in long-tailed learning, particularly methods leveraging multi-expert architectures and hypernetworks for controllable systems, and draws inspiration from the growing use of natural language for interacting with machine learning models.

▷ **Addressing Class Imbalance in Long-Tailed Learning.** The core challenge of long-tailed learning stems from severe class imbalance, where most labels have few samples. A significant body of work has focused on mitigating this. *Data-level approaches* attempt to balance class distributions through re-sampling, such as over-sampling minority classes (e.g., SMOTE Chawla et al. (2002)) or under-sampling majority ones Drummond et al. (2003). *Classifier-level adjustments* often involve re-weighting the loss function to emphasize tail classes Lin et al. (2017); Cui et al. (2019); Cao et al. (2019) or designing specialized loss functions that incorporate class priors or adjust logits, like Balanced Softmax Ren et al. (2020) or Logit Adjustment Zhao et al. (2022). Other strategies include *decoupled training*, where representation learning is separated from classifier learning to allow for tailored adjustments Kang et al. (2019); Zhou et al. (2020), and *knowledge transfer* techniques that aim to generalize from head to tail classes Yin et al. (2019); Liu et al. (2019). While these methods have improved performance on imbalanced datasets, they typically offer *a static solution or require manual tuning* for different preference trade-offs, unlike *the dynamic, user-guided IPE*.

▷ **Multi-Expert Systems and Hypernetwork-based Controllability.** To better handle the diverse learning needs across the spectrum of class popularities, *multi-expert architectures* have been developed. These systems typically train an ensemble of specialized classifiers, with each expert potentially focusing on different data regimes or class subsets Wang et al. (2020); Hong et al. (2021); Zhang et al. (2022). The outputs of these experts are then combined, often with sophisticated routing or gating mechanisms. Building on this, the concept of *controllable long-tailed learning* has emerged, enabling users to specify desired performance characteristics. A pivotal approach in this domain utilizes hypernetworks Ha et al. (2016) to generate the parameters of these expert classifiers conditioned on a numerical preference vector Zhao et al. (2024). This allows the creation of a diverse set of models along a Pareto front of head-tail performance. Our work directly leverages such a

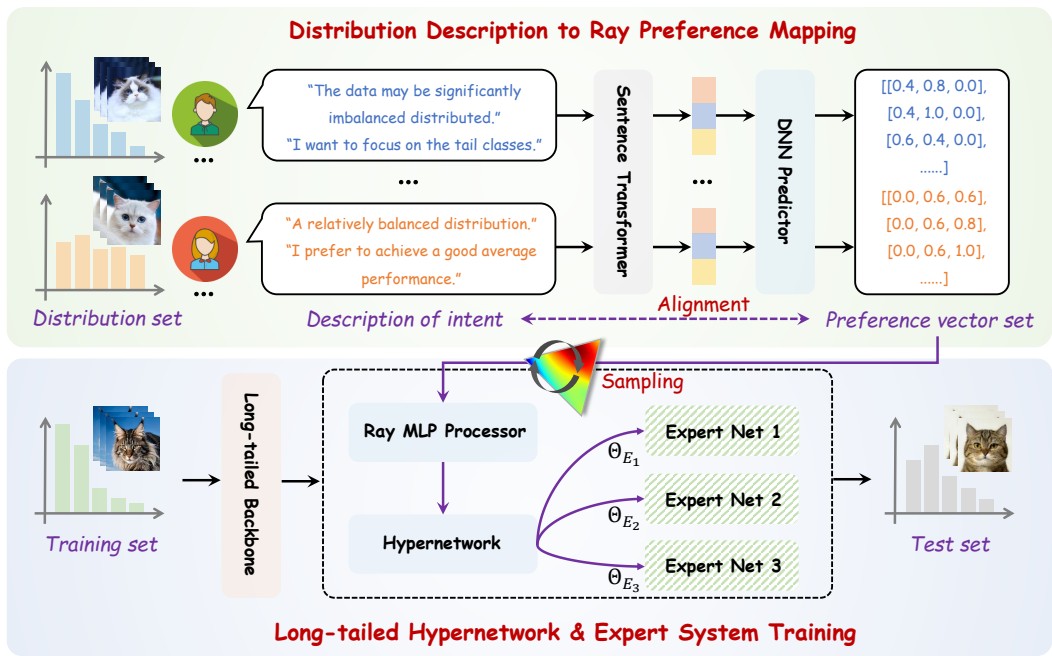

Figure 1: Framework for natural language-guided hypernetwork adaptation for preference-aware long-tailed learning. The upper part illustrates mapping user-expressed distributional preferences via natural language ("Description of intent") to low-dimensional "ray" preference vectors. This involves semantic encoding via a Sentence Transformer, followed by a DNN Predictor generating the preference vector, trained using empirically aligned pairs of (distribution characteristic, preference vector). The lower part depicts the core long-tailed learning system: a long-tailed backbone extracts features, and a hypernetwork (including a Ray MLP Processor), guided by the predicted preference vector, generates parameters ($\Theta_{E_1}, \Theta_{E_2}, \Theta_{E_3}$) for multiple expert networks to adapt to specific test set distributions.

hypernetwork-driven multi-expert system as its foundation, focusing on enhancing its accessibility. The underlying specialized loss in such systems, as conceptualized in Equation 3, implicitly navigates performance trade-offs by encouraging focus on more challenging expert components.

▷ **Natural Language for Interacting with ML Systems.** The use of natural language to configure, instruct, or interact with complex machine learning systems is a growing field aimed at improving usability and interpretability. This paradigm has seen applications in instruction-following agents Misra et al. (2017), interactive learning, and large vision-language models capable of responding to textual prompts Radford et al. (2021). While some research has explored natural language for specifying model architectures or training parameters, these often operate in constrained domains or require intricate semantic parsing.

Our contribution lies in synergizing these areas: we enhance a hypernetwork-controlled multi-expert long-tailed learning system by introducing a novel interface that translates natural language preferences into actionable control signals, thereby making sophisticated model adaptation more intuitive and accessible.

## 3 METHODOLOGY

This section details the Interpretable Preference Elicitation (IPE) framework. Our approach introduces a novel layer of abstraction that enables users to control a complex long-tailed learning model using natural language. We begin by establishing the preliminaries and formally defining our objective. We then present the architectural design of IPE, justifying our decoupled approach, and conclude with a granular description of each component and the specific algorithm for its training.

## 3.1 PRELIMINARIES AND PROBLEM FORMULATION

▷ **Parameterizable Long-Tailed Recognition Model.** We build upon a class of advanced long-tailed recognition systems whose behavior can be controlled at inference time. Let $f(\mathbf{x}; \Theta_{backbone}, \Theta_{experts})$ be such a model, comprising a main feature extractor (backbone) with parameters $\Theta_{backbone}$ and a set of $N_{exp}$ expert classifiers with parameters $\Theta_{experts}$. The key feature of this underlying system is a Hypernetwork, $\Phi_{HNet}$, which acts as a meta-model. The hypernetwork takes a $k$-dimensional ***numerical preference vector*** $r \in \mathcal{R} \subset \mathbb{R}^k$ as input and generates the parameters for the expert classifiers:

$$\Theta_{experts} = \Phi_{HNet}(r; \theta_H) \tag{1}$$

where $\theta_H$ are the pre-trained parameters of the hypernetwork. The vector $r = (r_1, \ldots, r_k)$ dictates the model's trade-off strategy, where each component $r_i$ might correspond to a preference for a certain subset of classes (e.g., head, medium, tail). While this provides control, it presents a significant ***usability gap***: a non-expert user cannot intuitively determine the optimal vector $r$ to match their specific, high-level intent.

▷ **Objective: Learning A Language-to-Preference Mapping.** Our primary objective is to bridge this usability gap by learning a mapping function, $\mathcal{F}_{IPE}$, that translates a user's intent, expressed as a natural language text description $\mathcal{D}_{\text{text}}$, into an effective numerical preference vector $\hat{r}$. This can be formally stated as:

$$\hat{r} = \mathcal{F}_{IPE}(\mathcal{D}_{\text{text}}) \tag{2}$$

The goal is to design and train $\mathcal{F}_{IPE}$ such that the resulting model, configured by $\Theta_{experts} = \Phi_{HNet}(\hat{r})$, exhibits performance characteristics that align with the semantic meaning of $\mathcal{D}_{\text{text}}$.

## 3.2 THE IPE FRAMEWORK: A DECOUPLED MAPPING ARCHITECTURE

To implement $\mathcal{F}_{IPE}$, we propose a decoupled, two-module architecture. A direct, end-to-end training that maps $\mathcal{D}_{\text{text}}$ to final model performance would involve an intractable joint optimization problem over the combined parameter spaces of the language model and the hypernetwork:

$$\min_{\theta_{IPE}, \theta_H} \mathbb{E}_{(\mathcal{D}_{\text{text}}, \mathcal{D}_{data})} \left[ \mathcal{L}_{\text{task}}(f(\mathbf{x}; \Theta_{backbone}, \Phi_{HNet}(\mathcal{F}_{IPE}(\mathcal{D}_{\text{text}}))), y) \right] \tag{3}$$

This approach is fraught with instability and optimization challenges. Instead, our IPE framework decouples the problem into two sequential, more manageable stages, assuming a pre-trained hypernetwork $\Phi_{HNet}$:

❶ *Preference Prediction*: A dedicated mapping module, which constitutes our core contribution, predicts a preference vector from text: $\hat{r} = \mathcal{F}_{IPE}(\mathcal{D}_{\text{text}})$.

❷ *Parameter Generation*: The pre-trained hypernetwork generates the expert parameters using this vector: $\Theta_{experts} = \Phi_{HNet}(\hat{r})$.

Our work focuses on the design and, crucially, the training of $\mathcal{F}_{IPE}$, which itself consists of two sub-modules: a semantic encoder and a preference predictor. The overall mapping is a composition:

$$\mathcal{F}_{IPE}(\mathcal{D}_{\text{text}}) = (\Phi_{PVP} \circ \Phi_{SE})(\mathcal{D}_{\text{text}}) = \Phi_{PVP}(\Phi_{SE}(\mathcal{D}_{\text{text}})) \tag{4}$$

## 3.3 COMPONENT ARCHITECTURE

▷ **Semantic Encoder ($\Phi_{SE}$):** This module is responsible for extracting the semantic essence from the user's input. We employ a pre-trained Sentence Transformer model, 'all-MiniLM-L6-v2', as $\Phi_{SE}$. It maps a variable-length text string $\mathcal{D}_{\text{text}}$ to a fixed-size, dense vector $e_{\text{text}} \in \mathbb{R}^{d_{emb}}$, where $d_{emb} = 384$.

$$\Phi_{SE} : \mathcal{S} \to \mathbb{R}^{d_{emb}}, \quad e_{\text{text}} = \Phi_{SE}(\mathcal{D}_{\text{text}}) \tag{5}$$

where $\mathcal{S}$ is the space of all possible input sentences.

▷ **Preference Vector Predictor ($\Phi_{PVP}$):** This module translates the general semantic embedding into the specific, low-dimensional preference space of the underlying model. It is a fully-connected Deep Neural Network (DNN), $\Phi_{PVP}(\cdot; \theta_{PVP})$, with parameters $\theta_{PVP}$.

$$\Phi_{PVP} : \mathbb{R}^{d_{emb}} \to \mathbb{R}^k \tag{6}$$

In our implementation, $k = 3$, corresponding to intuitive preferences for head, medium, and tail class performance. The network architecture consists of an input layer of size $384$, followed by 3 hidden layers of sizes $512$, $256$, and $128$ with ReLU activations, and a final linear output layer of size 3.

## 3.4 TRAINING ALGORITHM: SUPERVISED LEARNING VIA EMPIRICAL ANCHOR CALIBRATION

The key to making the IPE framework effective is to train the Preference Vector Predictor, $\Phi_{PVP}$, with high-quality supervision. We achieve this through a novel three-step process we term ***empirical anchor calibration***, which systematically creates a dataset linking natural language to optimal model configurations.

▷ **Step 1: Empirical Anchor Calibration.** This initial, offline stage establishes a ground-truth mapping between high-level distributional concepts and their corresponding optimal numerical preference vectors. This is the foundation of our supervised training.

❶ First, we define a set of $M$ canonical distributional scenarios, $\mathcal{C} = \{c_1, \ldots, c_M\}$. Each scenario $c_j$ represents a distinct and important long-tailed distribution type (e.g., 'forward50' for head-heavy, 'uniform', 'backward50' for tail-heavy).

❷ Next, we define a discrete grid of candidate preference vectors to search through, denoted as $\mathcal{R}_{grid}$. This grid is formed by sampling values for each of the $k$ dimensions of a vector $r$, for instance, $\mathcal{R}_{grid} = \{r \mid r_i \in \{0.0, 0.1, \ldots, 1.0\}\}_{i=1}^{k}$. This allows for a systematic exploration of the preference space.

❸ For each scenario $c_j \in \mathcal{C}$, we perform an exhaustive evaluation. We take our pre-trained long-tailed model, $f(\mathbf{x}; \Theta_{backbone}, \Phi_{HNet}(r))$, and run it on the test dataset $\mathcal{D}_{test}^{(j)}$ that corresponds to scenario $c_j$. This is done for every single candidate vector $r$ in our search grid $\mathcal{R}_{grid}$.

❹ Finally, for each scenario $c_j$, we identify the set of preference vectors that yield the best performance (e.g., highest top-1 accuracy). This set is designated as the ***Optimal Anchor Vector Set***, $\mathcal{R}_j^*$. Formally, this is expressed as:

$$\mathcal{R}_j^* = \arg \underset{r \in \mathcal{R}_{grid}}{\text{top-K}} \left( \text{EvalMetric}(f(\cdot; \Theta_{backbone}, \Phi_{HNet}(r)), \mathcal{D}_{test}^{(j)}) \right) \tag{7}$$

Here, EvalMetric is the evaluation function (e.g., accuracy), and arg top-K selects the K best-performing vectors $r$ from the grid $\mathcal{R}_{grid}$. These empirically validated anchor sets, $\mathcal{R}_j^*$, now serve as high-quality "ground-truth" targets for our supervised learning task.

▷ **Step 2: Corpus Generation.** With the numerical targets established, we now generate the corresponding natural language inputs. For each canonical scenario $c_j$, we use a large language model to generate a large and diverse set of textual descriptions, $\mathcal{T}_j$. Each text in $\mathcal{T}_j$ semantically describes the concept of $c_j$. For example, for the 'backward50' scenario, a generated text could be "strong emphasis on rare classes" or "prioritize performance on underrepresented categories." This diversity ensures the final model is robust to different phrasings of the same intent.

▷ **Step 3: Supervised Training of the Predictor.** In the final step, we construct the training dataset $\mathcal{D}_{train}$ and train our predictor network $\Phi_{PVP}$. We create pairs by linking each text description to a corresponding optimal anchor vector:

$$\mathcal{D}_{train} = \bigcup_{j=1}^{M} \{(\mathcal{D}_{text}, r^*) \mid \mathcal{D}_{text} \in \mathcal{T}_j, r^* \in \mathcal{R}_j^*\} \tag{8}$$

This equation means that for every scenario $j$ (from 1 to $M$), we pair each of its textual descriptions ($\mathcal{D}_{text} \in \mathcal{T}_j$) with an optimal preference vector ($r^*$) randomly sampled from its corresponding anchor set $\mathcal{R}_j^*$. The union symbol $\bigcup$ indicates that we combine these pairs from all scenarios into a single large training dataset, $\mathcal{D}_{train}$.

With this dataset, we train the parameters $\theta_{PVP}$ of the predictor network $\Phi_{PVP}$ by minimizing the Mean Squared Error (MSE) between the predicted vectors and the ground-truth anchor vectors:

$$\min_{\theta_{PVP}} \frac{1}{|\mathcal{D}_{train}|} \sum_{(\mathcal{D}_{text}, r^*) \in \mathcal{D}_{train}} \|\Phi_{PVP}(\Phi_{SE}(\mathcal{D}_{text})) - r^*\|_2^2 \tag{9}$$

In this loss function, for each text-vector pair $(\mathcal{D}_{text}, r^*)$ in the training set:

- $\Phi_{SE}(\mathcal{D}_{\text{text}})$ first converts the raw text into a numerical semantic embedding.
- $\Phi_{PVP}(\cdot)$ then takes this embedding and predicts a preference vector.
- The term $\| \cdot - r^* \|_2^2$ calculates the squared Euclidean distance (error) between this predicted vector and the empirically-validated optimal anchor vector $r^*$.
- The overall objective is to adjust the predictor's parameters $\theta_{PVP}$ to make this error as small as possible on average over the entire training set.

This rigorously supervised process ensures that the IPE framework learns a precise and effective mapping from user language to optimal model control.

## 4 EXPERIMENTS

We validate IPE through comprehensive experiments and our evaluation focuses on demonstrating:

❶ The ability to effectively control head-tail trade-offs via natural language preferences.

❷ Superior or competitive performance compared to state-of-the-art (SOTA) long-tailed recognition methods under various test distributions, including uniform and shifted ones.

❸ The robustness and generalizability of our approach across diverse benchmark datasets. We first detail the experimental setup, followed by a presentation and discussion of the results.

### 4.1 EXPERIMENTAL SETUP

The experimental setup is crucial for understanding the context of our results. We meticulously define datasets, baselines, implementation specifics, and evaluation protocols to ensure reproducibility and fair comparison. Figure 2 provides an initial glimpse into our method's capability, showing how diverse textual inputs are mapped to distinct regions in the preference vector space, a foundational aspect for achieving interpretable control. This mapping is key to translating user intent into actionable model configurations. The subsequent combined figure (Figure 3) will further detail performance aspects and model behavior related to these preferences.

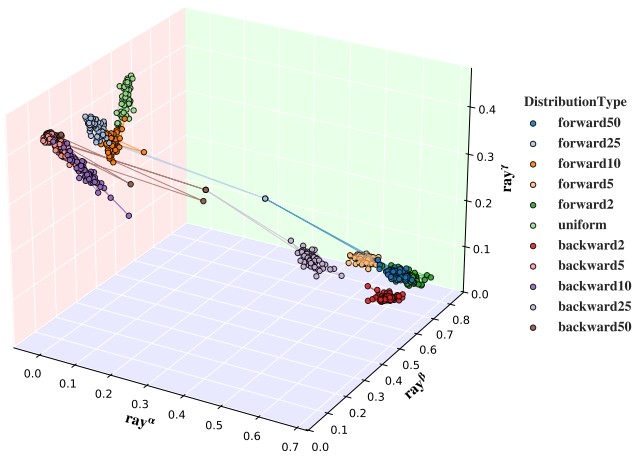

Figure 2: Distribution of learned preference vectors $r$ for different textual inputs, illustrating the mapping diversity. Different colors correspond to preference vectors optimized for distinct canonical distribution types (e.g., forward-LT, uniform, backward-LT).

▷ **Datasets.** Our approach is evaluated on three standard long-tailed benchmarks: CIFAR-100-LT Cao et al. (2019), ImageNet-LT Liu et al. (2019), and iNaturalist 2018 Van Horn et al. (2018), which exhibit diverse class imbalance ratios (IR $\rho$) from 100 to 500. CIFAR-100-LT is generated from CIFAR-100 Krizhevsky et al. (2009) by inducing an exponential decay in sample counts across its 100 classes, and we evaluate on IRs $\rho \in \{10, 50, 100\}$ following prior work Cao et al. (2019). ImageNet-LT, a subset of ImageNet-2012 Deng et al. (2009), comprises $1,000$ classes (115.8K training images) with sample counts following a Pareto distribution ($\alpha = 6$), resulting in $\rho = 256$. iNaturalist 2018 is a large-scale real-world dataset featuring $\sim$437K images across $8,142$ species, with a natural long-tail distribution yielding $\rho \approx 500$.

▷ **Baselines.** We compare IPE against a diverse set of SOTA long-tailed recognition methods: MiSLAS Zhong et al. (2021), RIDE Wang et al. (2020), SADE Zhang et al. (2022), Balanced Softmax (BS) Ren et al. (2020), LADE Hong et al. (2021), Causal Tang et al. (2020), LSC Wei et al. (2024),

Table 1: Control of head-tail trade-off using different preference vectors $R_i$ on CIFAR100-LT (IR=100). Values are Top-1 accuracy (%). Bold indicates the best performance for that specific test distribution among the $R_i$ vectors. The vectors $R_1$ to $R_5$ are chosen to represent preferences from strong head-class focus to strong tail-class focus.

| Test Distribution | IR | $R_1(0.0, 0.9, 0.3)$ | $R_2(0.1, 0.4, 0.6)$ | $R_3(0.3, 0.7, 0.2)$ | $R_4(0.6, 0.5, 0.7)$ | $R_5(0.7, 1.0, 0.5)$ |
|---|---|---|---|---|---|---|
| Forward | 50 | **70.3** | 69.2 | 69.0 | 68.8 | 69.0 |
|  | 25 | 65.6 | **66.8** | 65.7 | 65.4 | 65.5 |
| Uniform | 1 | 51.9 | 51.8 | **52.3** | 52.0 | 51.8 |
| Backward | 25 | 45.2 | 45.4 | 45.7 | **46.9** | 45.9 |
|  | 50 | 46.6 | 47.1 | 47.6 | 47.5 | **48.8** |

BalPoE Aimar et al. (2023), PRL Zhao et al. (2024), and SSE-SAM Lyu et al. (2025). Further details on these baselines are provided in Appendix.

▷ **Implementation Details.** Standard backbones are used: ResNeXt-50 for ImageNet-LT, ResNet-32 for CIFAR100-LT, and ResNet-50 for iNaturalist 2018. Hypernetworks (MLPs) generate expert parameters, with a cosine classifier for predictions. Key hyperparameters include: Dirichlet $\alpha = 1.2$, stochastic annealing $\mu = 0.3$, SGD (momentum 0.9), 200 epochs, initial learning rate 0.1 (with linear decay). The Text-to-Preference-Vector Predictor is trained for 200 epochs (batch size 128) with KL divergence as loss function. Full implementation details are available in Appendix.

▷ **Evaluation Protocol.** Following Hong et al. (2021); Zhang et al. (2022), we evaluate using micro accuracy on multiple test datasets reflecting 11 different class imbalance levels. These range from forward long-tailed (e.g., IR = 50) to uniform, and further to backward long-tailed (e.g., IR = 0.02, calculated as $\min N_i / \max N_i$ for reverse scenarios). We also report accuracy on many-shot ($> 100$ samples), medium-shot (20-100 samples), and few-shot ($< 20$ samples) class groups. Confidence intervals are derived from five runs. Appendix provides details on test distribution generation.

## 4.2 Results and Analysis

▷ **Controllability and Preference Mapping.** A core tenet of IPE is the effective translation of natural language into control signals. We verify this both quantitatively and qualitatively. Table 1 demonstrates precise, quantitative control: a control vector tailored for head classes ($R_1$) achieves peak performance (70.3%) on a head-heavy (Forward-50) distribution, while a vector favoring tail classes ($R_5$) excels (48.8%) on a tail-heavy (Backward-50) set, showcasing fine-grained control across the performance spectrum. This effective control is underpinned by a well-structured semantic mapping, as visualized in Figure 2 and Figures 3c-3d. These figures show that textual directives corresponding to different distribution types form distinct, well-separated clusters in the control space, confirming that IPE learns a robust and meaningful translation from user intent to specific model behaviors.

▷ **Performance on Uniform Test Distributions.** Table 2 details the Top-1 accuracy of IPE compared to SOTA baselines when the test class distribution is uniform. IPE consistently demonstrates strong performance, often surpassing existing methods. For example, on CIFAR100-LT with IR=100, IPE achieves 52.3% accuracy, outperforming PRL (52.2%) and SADE (48.8%). On the large-scale iNaturalist 2018 dataset, IPE (75.5%) shows an improvement over BalPoE (75.0%) and PRL (75.1%). Similarly, for ImageNet-LT, IPE (61.1%) slightly improves upon PRL (60.8%). These results indicate that the enhanced interpretability and control offered by IPE do not come at the cost of performance and can even lead to modest gains on standard balanced evaluation settings.

▷ **Performance on Unknown (Shifted) Test Distributions.** IPE's robustness to distribution shifts, without prior knowledge of the test distribution (✗), is evaluated on CIFAR100-LT (IR=100) across 11 diverse test distributions, as shown in Table 3. IPE consistently achieves leading performance, particularly in challenging backward long-tailed scenarios which significantly deviate from the training distribution. For instance, on the Backward-LT IR=50 test set (most extreme reverse tail), IPE (48.8%) outperforms the strong baseline PRL (48.5%).

Similar strong performance trends are observed for ImageNet-LT and iNaturalist 2018. On ImageNet-LT dataset (Table 4), our IPE approach demonstrates unmatched Top-1 accuracy. Notably, in

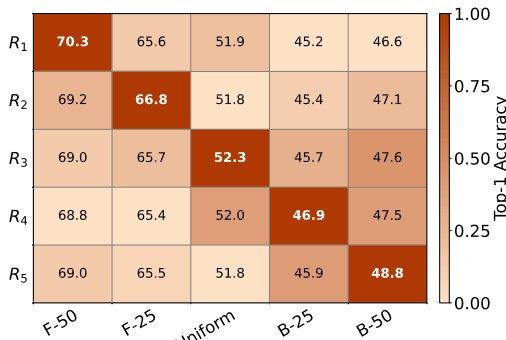

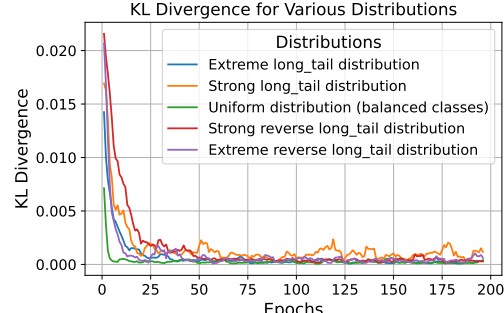

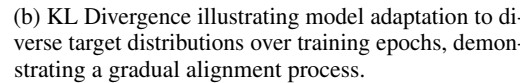

(a) Heatmap of Top-1 accuracy for different preference vectors $R_i$ in various test distributions (F=Forward-LT, B=Backward-LT, IR indicated by number).

(b) KL Divergence illustrating model adaptation to diverse target distributions over training epochs, demonstrating a gradual alignment process.

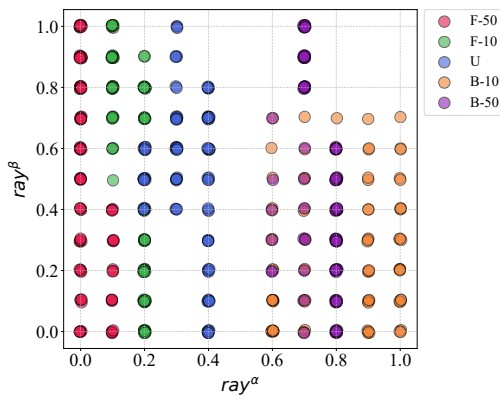

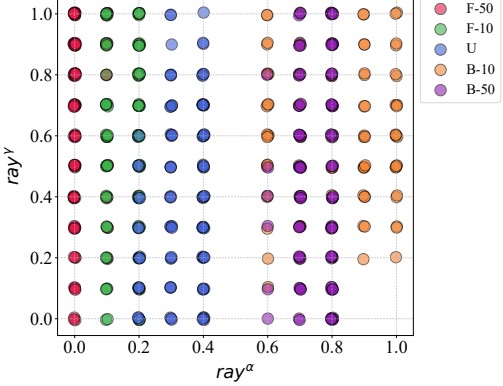

(c) Projection of learned preference vectors onto the $(r_x, r_y)$ plane, showing distinct clusters corresponding to different distribution types.

(d) Alternate projection of learned preference vectors onto the $(r_x, r_z)$ plane, further illustrating cluster separation.

Figure 3: Performance analysis and preference space visualizations. (a) Demonstrates the controllability of performance trade-offs by applying different preference vectors. (b) Analyzes the model's adaptation dynamics to target distributions. (c-d) Visualize the structured nature of the learned preference space from different 2D perspectives.

Table 2: Top-1 accuracy (%) on CIFAR100-LT, iNaturalist 2018, and ImageNet-LT when the test class distribution is uniform. IPE ("Ours") is compared against SOTA methods.

| Method | CIFAR100-LT | | | iNaturalist 2018 | ImageNet-LT |
|---|---|---|---|---|---|
| | IR=10 | IR=50 | IR=100 | | |
| Softmax | 59.1 | 45.6 | 41.4 | 64.7 | 48.0 |
| Causal Tang et al. (2020) | 59.4 | 48.8 | 45.0 | 64.4 | 50.3 |
| BS Ren et al. (2020) | 61.0 | 50.9 | 46.1 | 70.6 | 52.3 |
| MiSLAS Zhong et al. (2021) | 62.5 | 51.5 | 46.8 | 70.7 | 51.4 |
| LADE Hong et al. (2021) | 61.6 | 50.1 | 45.6 | 69.3 | 52.3 |
| RIDE Wang et al. (2020) | 61.8 | 51.7 | 48.0 | 71.8 | 56.3 |
| SADE Zhang et al. (2022) | 63.6 | 53.8 | 48.8 | 72.7 | 58.8 |
| LSC Wei et al. (2024) | 65.0 | 56.5 | 51.8 | 73.9 | 60.2 |
| BalPoE Aimar et al. (2023) | 64.8 | 56.3 | 52.0 | 75.0 | 59.3 |
| PRL Zhao et al. (2024) | 65.6 | 57.3 | 52.2 | 75.1 | 60.8 |
| SSE-SAM Lyu et al. (2025) | 64.5 | 55.6 | 49.3 | 69.8 | 47.5 |
| **Ours (IPE)** | **66.2** | **57.8** | **52.3** | **75.5** | **61.1** |

Table 3: Top-1 accuracy (%) on CIFAR100-LT (IR=100) with various unknown test class distributions. *Prior*: ✓indicates test class distribution is known and used by the method at test time, ✗otherwise. IPE ("Ours") operates without prior knowledge.

| Method | Prior | Forward-LT (IR) | | | | | Uni | Backward-LT (IR) | | | | |
|---|---|---|---|---|---|---|---|---|---|---|---|---|
| | | 50 | 25 | 10 | 5 | 2 | 1 | 2 | 5 | 10 | 25 | 50 |
| Softmax | ✗ | 63.3 | 62.0 | 56.2 | 52.5 | 46.4 | 41.4 | 36.5 | 30.5 | 25.8 | 21.7 | 17.5 |
| BS | ✗ | 57.8 | 55.5 | 54.2 | 52.0 | 48.4 | 46.1 | 43.6 | 40.8 | 38.4 | 36.3 | 33.7 |
| MiSLAS | ✗ | 58.8 | 57.2 | 55.2 | 53.0 | 49.6 | 46.8 | 43.6 | 40.1 | 37.7 | 33.9 | 32.1 |
| LADE | ✗ | 56.0 | 55.5 | 52.8 | 51.0 | 48.0 | 45.6 | 43.2 | 40.0 | 38.3 | 35.5 | 34.0 |
| LADE | ✓ | 62.6 | 60.2 | 55.6 | 52.7 | 48.2 | 45.6 | 43.8 | 41.1 | 41.5 | 40.7 | 41.6 |
| RIDE | ✗ | 63.0 | 59.9 | 57.0 | 53.6 | 49.4 | 48.0 | 42.5 | 38.1 | 35.4 | 31.6 | 29.2 |
| SADE | ✗ | 65.2 | 62.5 | 58.8 | 55.4 | 51.2 | 48.8 | 43.0 | 43.9 | 42.4 | 42.2 | 42.0 |
| LSC | ✗ | 67.8 | 64.2 | 60.2 | 58.1 | 53.2 | 51.6 | 44.7 | 45.7 | 44.2 | 44.7 | 48.0 |
| BalPoE | ✗ | 69.0 | 65.2 | 61.2 | 59.0 | 54.2 | 51.7 | 45.7 | 46.6 | 45.2 | 45.2 | 45.8 |
| PRL | ✗ | 69.5 | 65.7 | 61.7 | 59.3 | 54.7 | 52.2 | 46.2 | 47.1 | 45.7 | 45.7 | 48.5 |
| **Ours (IPE)** | ✗ | **70.3** | **66.8** | **62.5** | **59.4** | **55.4** | **52.3** | **49.5** | **47.6** | **46.9** | **46.9** | **48.8** |

Table 4: Top-1 accuracy (%) on ImageNet-LT with various unknown test class distributions.

| Method | Prior | Forward-LT (IR) | | | | | Uni | Backward-LT (IR) | | | | |
|---|---|---|---|---|---|---|---|---|---|---|---|---|
| | | 50 | 25 | 10 | 5 | 2 | 1 | 2 | 5 | 10 | 25 | 50 |
| Softmax | ✗ | 66.1 | 63.8 | 60.3 | 56.6 | 52.0 | 48.0 | 43.9 | 38.6 | 34.9 | 30.9 | 27.6 |
| BS | ✗ | 63.2 | 61.9 | 59.5 | 57.2 | 54.4 | 52.3 | 50.0 | 47.0 | 45.0 | 42.3 | 40.8 |
| MiSLAS | ✗ | 61.6 | 60.4 | 58.0 | 56.3 | 53.7 | 51.4 | 49.2 | 46.1 | 44.0 | 41.5 | 39.5 |
| LADE | ✗ | 63.4 | 62.1 | 59.9 | 57.4 | 54.6 | 52.3 | 49.9 | 46.8 | 44.9 | 42.7 | 40.7 |
| LADE | ✓ | 65.8 | 63.8 | 60.6 | 57.5 | 54.5 | 52.3 | 50.4 | 48.8 | 48.6 | 49.0 | 49.2 |
| RIDE | ✗ | 67.6 | 66.3 | 64.0 | 61.7 | 58.9 | 56.3 | 54.0 | 51.0 | 48.7 | 46.2 | 44.0 |
| SADE | ✗ | 69.7 | 67.5 | 65.4 | 62.3 | 60.3 | 58.3 | 56.7 | 54.9 | 54.3 | 53.1 | 52.6 |
| LSC | ✗ | 72.0 | 69.7 | 67.5 | 65.3 | 62.7 | 60.2 | 59.2 | 58.5 | 57.9 | 57.5 | 57.0 |
| BalPoE | ✗ | 72.2 | 69.7 | 67.2 | 64.3 | 62.2 | 59.5 | 58.5 | 57.7 | 56.9 | 56.7 | 56.6 |
| PRL | ✗ | 72.7 | 70.2 | 68.0 | 65.8 | 63.2 | 60.7 | 59.7 | 59.0 | 58.4 | 58.0 | 57.5 |
| **Ours (IPE)** | ✗ | **73.6** | **71.4** | **69.1** | **66.2** | **63.9** | **61.0** | **60.2** | **59.8** | **59.1** | **58.9** | **57.8** |

Forward-LT configurations, as the ratio of unknown classes declines from 50% to 2%, IPE's Top-1 accuracy decreases from 73.6% to 63.9%, while still maintaining a significant lead over all baseline methodologies. In Backward-LT scenarios, IPE's accuracy progressively rises from 57.8% to 60.2%, outperforming all comparative approaches. See Appendix for the analysis on iNaturalist 2018.

These findings comprehensively illustrate the method's exceptional performance and resilience in managing diverse unknown class distributions. The KL divergence analysis (Figure 3b, part of Figure 3b), which shows the underlying model's rapid convergence when adapting to various target distributions, further underscores the strong adaptive capabilities harnessed and guided by IPE.

## 5 CONCLUSION

In this work, we introduced a novel framework enabling users to guide controllable long-tailed learning systems via natural language. By translating textual descriptions of desired performance trade-offs into low-dimensional preference vectors, our method seamlessly integrates intuitive user commands with a hypernetwork-based expert system. Our approach significantly enhances the usability and interpretability of preference-adaptive long-tailed models without compromising their adaptability or performance. By bridging the gap between high-level user intent and concrete model configurations, this work paves the way for more accessible and practical long-tailed learning systems capable of catering to diverse real-world demands and user-specific needs.

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

# Appendix
# Interpretable Preference Elicitation:
# Aligning User Intent with Controllable Long-tailed Learning

## A  BASELINES DETAILS

In this section, we present a detailed review of several state-of-the-art long-tailed recognition methods, which are used as baselines to evaluate the effectiveness of our proposed IPE method.

- **Two-stage methods** These approaches aim to reduce the bias toward head classes by decoupling feature representation learning from classifier training. MiSLAS Zhong et al. (2021) enhances tail class learning by incorporating a mixup-based technique during the second training stage. This separation helps minimize the adverse effects of data imbalance on feature learning.

- **Logit-adjusted training approaches** tackle class imbalance by altering the logits during model training. Balanced Softmax Ren et al. (2020) incorporates a frequency-aware term into the softmax formulation, enabling adaptive logit scaling based on class distribution. LADE Hong et al. (2021) further separates representation learning from classifier optimization by introducing a learnable logit correction. These strategies help mitigate the dominance of head classes and promote more balanced predictions.

- **Ensemble learning methods** aim to address data imbalance by utilizing multiple classifiers or expert models to capture data diversity. RIDE Wang et al. (2020) trains several experts using distinct resampling techniques and adaptively aggregates their predictions based on the underlying sample distribution. Building on this idea, SADE Zhang et al. (2022) introduces a self-adaptive knowledge distillation framework to facilitate information sharing among experts. By harnessing the complementary strengths of different experts, these methods offer improved robustness in long-tailed scenarios.

- **Causal inference-based methods** tackle the long-tailed recognition challenge by developing classifiers grounded in causal reasoning. Causal Tang et al. (2020) introduces a framework that estimates the causal impact of each class on the model's predictions, effectively mitigating the bias caused by imbalanced class distributions.

- **Representation learning methods** address long-tailed recognition by aiming to learn feature representations that are both balanced and highly discriminative. LSC Wei et al. (2024) proposes a contrastive learning framework that jointly considers instance-level and group-level distribution alignment, thereby enhancing the feature quality, especially for tail classes.

- **Balanced posterior averaging methods** aim to aggregate the outputs of multiple experts by weighting their predictions according to posterior probabilities. BalPoE Aimar et al. (2023) introduces a strategy that prioritizes experts with stronger performance on tail classes, enabling a more effective balance between head and tail class predictions.

- **Controllable Expert Ensemble Methods** This approach addresses distribution shift and performance trade-offs in long-tailed recognition. PRL Zhao et al. (2024) generates a diverse set of experts using a hypernetwork and learns preference-aware combinations through Pareto optimization. At test time, it allows flexible adjustment of predictions based on user-defined preferences without retraining. PRL offers stronger adaptability, interpretability, and generalization across diverse test distributions.

Although many existing methods have made progress in handling long-tailed distributions, they often depend on predefined assumptions about data during training or inference, which limits their generalizability in practical scenarios. Furthermore, most approaches lack the flexibility to adjust the performance balance between head and tail classes based on user intent. To address this, we introduce IPE—a novel method that allows users to influence model behavior through a simple textual description of the test distribution. By aligning these descriptions with a set of learned preference vectors, IPE enables the expert ensemble to automatically adapt to a wide range of unseen distributions, without requiring distribution-specific retraining or prior knowledge.

Table 5: The main parameter settings for DNN model training

| Parameter | Value | Parameter | Value |
|---|---|---|---|
| Model architecture | 4-layer fully connected DNN | Input dimension | 384 |
| Train/test split | 70% / 30% | Output dimension | 3 |
| Hidden layer sizes | [512, 256, 128] | Activation function | ReLU |
| Optimizer | Adam | Loss function | KL divergence |
| Learning rate | 0.001 | Epochs | 200 |
| Text encoder | all-MiniLM-L6-v2 | Batch size | 128 |

## B  IMPLEMENTATION DETAILS

We evaluate the generalization and robustness of our proposed IPE method on three representative long-tailed benchmarks: CIFAR-100-LT, ImageNet-LT, and iNaturalist 2018. Implementation details are tailored to the specific characteristics of each dataset.

For CIFAR-100-LT, following the protocol Cao et al. (2019), we use ResNet-32 as the backbone and train under three imbalance ratios: {10, 50, 100}. The model is trained for 200 epochs using SGD with a momentum of 0.9, a batch size of 128, and an initial learning rate of 0.1. A linear decay schedule is applied for learning rate adjustment.

For the two large-scale datasets, we adopt settings aligned with prior works Wang et al. (2020); Zhang et al. (2022). On ImageNet-LT, we employ ResNeXt-50 Xie et al. (2017) as the backbone and train for 180 epochs with an imbalance ratio of 256. The SGD optimizer is used with a momentum of 0.9, a batch size of 64, and an initial learning rate of 0.025, using cosine annealing Loshchilov & Hutter (2016) as the learning rate scheduler.On iNaturalist 2018, we use a ResNet-50 backbone, train for 100 epochs, and set the batch size to 512. The imbalance ratio is 500, and we adopt the same SGD optimizer configuration with an initial learning rate of 0.2, also using cosine annealing for scheduling.

In addition, for the hypernetwork module, we adopt shared settings across all experiments: Dirichlet parameter $\alpha = 1.2$ and stochastic annealing coefficient $\mu = 0.3$.

For the Text-to-Preference-Vector Predictor, the main training parameters are detailed in Table 5.

## C  RESULTS ON INATURALIST 2018 DATASETS

On the iNaturalist 2018 dataset, IPE continues to exhibit remarkable performance. Within Forward-LT setups, as the unknown class proportion reduces from 3 to 2, IPE's Top-1 accuracy experiences a marginal increase from 74.2% to 74.3%, reaching an optimal performance of 75.1% under uniform distribution. In Backward-LT configurations, despite a slight accuracy decrease from 74.9% to 74.3%, IPE consistently outperforms all comparative methods. These results further solidify the method's broad applicability across distinct datasets and experimental contexts.

In summary, by effectively addressing the challenges posed by long-tailed and unknown class distributions, the IPE method showcases superior performance on two representative long-tailed datasets, thereby confirming its methodological superiority.

## D  DATASETS

To comprehensively assess the effectiveness of our proposed approach, we perform evaluations on three representative long-tailed benchmarks: CIFAR100-LT, ImageNet-LT, and iNaturalist 2018. These datasets span diverse domains and possess varying imbalance severities, making them well-suited for validating long-tailed recognition methods.

**CIFAR100-LT**  Cao et al. (2019) This dataset is derived from CIFAR-100 by introducing a long-tailed distribution, where the number of samples per class decreases exponentially. It consists of 60,000 color images, each with a resolution of $32 \times 32$ pixels, spanning 100 distinct classes, and supports imbalance ratios as high as 100.

Table 6: Top-1 accuracy on iNaturalist 2018 with various unknown test class distributions.

| Method | Prior | Forward-LT | | Uni. | Backward-LT | |
|---|---|---|---|---|---|---|
| | | 3 | 2 | 1 | 2 | 3 |
| Softmax | ✗ | 65.4 | 65.5 | 64.7 | 64.0 | 63.4 |
| BS | ✗ | 70.3 | 70.5 | 70.6 | 70.6 | 70.8 |
| MiSLAS | ✗ | 70.8 | 70.8 | 70.7 | 70.7 | 70.2 |
| LADE | ✗ | 68.4 | 69.0 | 69.3 | 69.6 | 69.5 |
| LADE | ✓ | - | 69.1 | 69.3 | 70.2 | - |
| RIDE | ✗ | 71.5 | 71.9 | 71.8 | 71.9 | 71.8 |
| SADE | ✗ | 72.3 | 72.6 | 72.7 | 73.0 | 73.2 |
| LSC | ✓ | - | - | - | - | - |
| BalPoE | ✗ | 73.1 | 73.5 | 73.8 | 73.6 | 73.5 |
| PRL | ✓ | 73.7 | 73.8 | 74.3 | 74.0 | 73.9 |
| **Ours** | ✗ | **74.2** | **74.3** | **75.1** | **74.9** | **74.3** |

Table 7: Details of the long-tailed datasets.

| Dataset | # Train | # Test | # Classes | Imbalance Ratio |
|---|---|---|---|---|
| CIFAR100-LT | 50,000 | 10,000 | 100 | {10, 50, 100} |
| ImageNet-LT | 115,846 | 50,000 | 1,000 | 256 |
| iNaturalist 2018 | 437,513 | 24,426 | 8,142 | 500 |

**ImageNet-LT** Liu et al. (2019) This long-tailed variant of the ImageNet dataset includes more than 115,000 images distributed across 1,000 categories. The class frequencies follow a Pareto distribution with a shape parameter of $\alpha = 6$, resulting in an imbalance ratio reaching up to 256.

**iNaturalist 2018** Van Horn et al. (2018) This real-world dataset exhibits an inherent long-tailed distribution, containing around 450,000 images covering 8,142 species. The sample counts per class vary widely, resulting in an imbalance ratio as high as 500, which makes the dataset particularly challenging due to both severe class imbalance and substantial intra-class diversity.

## E  LIMITATIONS

Although the proposed IPE framework demonstrates strong performance and interpretability, it still has certain limitations. First, the mapping from text to preference vectors heavily depends on the diversity and quality of the training corpus, which may limit the model's generalization to novel or nuanced user intents. Second, the current use of a three-dimensional preference space may constrain the granularity of control in complex scenarios. While increasing the dimensionality could enhance expressiveness, it may also introduce challenges in training stability and risk of overfitting.

## F  BROADER IMPACTS

The proposed Interpretable Preference Elicitation (IPE) method offers a practical solution to controllability and distribution adaptability in long-tailed learning. By leveraging semantically guided preference vectors, IPE enables users to dynamically adjust the trade-off between head and tail class performance without additional training, thus improving the flexibility and usability of the model.

The effectiveness and robustness of this approach have been validated across multiple long-tailed visual benchmarks, demonstrating strong potential for real-world applications such as medical diagnosis, ecological monitoring, and public safety, where accurate tail-class recognition is critical. Moreover, the built-in interpretability mechanism enhances user understanding and trust in model be-

havior, contributing to the development of more transparent, fair, and efficient long-tailed recognition systems.

## G   USE OF LLMS

We used LLM for writing polish to improve readability.

