# OpenReview forum: "Interpretable Preference Elicitation: Aligning User Intent with Controllable Long-tailed Learning"
_ICLR.cc/2026/Conference — ICLR 2026 Conference Withdrawn Submission_

### Official Review · Reviewer_dRpV · 2025-10-30

**Soundness:** 3
**Presentation:** 3
**Contribution:** 3
**Rating:** 4
**Confidence:** 4

**Summary:**

This paper proposes a novel framework - Interpretable Preference Elicitation which allows Mixture-of-Experts model predictions to be steered using natural language descriptions. The inference pipeline works using a two-staged approach - the natural language text is converted into vectors. The vectors serves as the routing or gating vectors for the MoE models. During training time, the authors conduct an extensive grid search over the gating vectors along the dimensions of steer-ability. Then they use an LLM to generate synthetic data which could be mapped to the vectors. The synthetic data is used to train a Sentence Transformer with a shallow MLP to map the descriptions into the vectors.

The authors apply this method to examine if the descriptions could encourage the model to improve the accuracy for long-tail classes. Through empirical studies, they show that this method shows better performance than several baselines

**Strengths:**

1. The motivation behind this paper is clear. The presentation with the figures makes the overall design easy to follow intuitively.
2. The empirical results show that this framework has better performance in comparison to several baselines for the vision benchmarks

**Weaknesses:**

1. The offline grid search seems to be a computationally intensive process and might not scale with more number of experts, dimensions and modes of steerability.
2. The central contribution of this paper is that the framework allows steerability through natural language. However, it is not clear whether this claim is proven without any user studies. In fact it is not clear how the natural language text is generated for the experiments to steer the model predictions. It is uncertain if this work could generalize to unforeseen texts.

**Questions:**

1. Why is a 3 dimensional preference vector chosen?

---

### Official Review · Reviewer_fGM5 · 2025-10-31

**Soundness:** 2
**Presentation:** 1
**Contribution:** 2
**Rating:** 0
**Confidence:** 2

**Summary:**

The paper develops a method to tailor neural networks to prediction tasks with long-tailed class distributions. The main novel contribution is a method to describe class trade-offs in natural language.

**Strengths:**

The task of specifying a tradeoff between common and tail classes in natural language is interesting. I'm not familiar with the related work, but I'm not sure if prior work has tried to do this.

**Weaknesses:**

1. Severe clarity issues:
- From the outset of the paper's abstract and introduction, it's unclear what problem they are studying and why.
- The methodology section is very hard to read - the authors do not clarify which part is novel and which part isn't, and there's a lot of notation introduced that ends up being confusing rather than clarifying.
- Minor: The related work section is all over the place. Inexplicably, it cites the GPT-3 paper, which I'm not sure of the relevance to this paper.
2. I do not understand the authors' choice of evaluation tasks. Supposedly, the novel contribution is the ability to specify desired handling of common classes vs. long-tail classes in natural language. But there are no metrics or evals for this capability. Instead, the authors just compare on standard metrics and datasets for long-tail prediction.
3. There are no examples of the natural language specifications in the paper, making it very hard to understand what the method is actually trying to achieve.
4. Related to the clarity concern in (1), the paper is not self-contained and I had to look at three other prior papers to even understand what problem this paper is studying.

Overall, because this paper is difficult-to-read, poorly motivated, and does not justify its evals, it should be a clear reject.

**Questions:**

1. In Table 3 and 4 you report SOTA against all prior long tail methods. What does this have to do with the paper's original motivation on describing preferences over head-tail tradeoffs in natural language?
2. What are example behaviors that your method allows (by specifying in natural language) that prior methods do not allow?

---

### Official Review · Reviewer_DK75 · 2025-10-31

**Soundness:** 3
**Presentation:** 4
**Contribution:** 2
**Rating:** 4
**Confidence:** 3

**Summary:**

This paper explores how to improve the usability of controllable hypernetworks by allowing users to directly specify the desired trade-offs or preferences through natural language. The proposed framework, Interpretable Preference Elicitation (IPE), follows a two-step training process: it first identifies canonical distributional scenarios with their corresponding optimal numerical preference vectors, and then associates these vectors with high-level textual descriptions. The paper demonstrates that this approach enhances usability and interpretability without degrading performance—and in some cases, even achieves slight improvements.

**Strengths:**

1. **Relevant direction.** Improving the usability of advanced methods such as preference-controlled hypernetworks is an important research direction toward democratizing machine learning.
2. **Clarity and simplicity.** The paper is clearly written, and the proposed approach is simple yet effective, substantially improving user-friendliness and interpretability.
3. **Structured methodology.** Presents a well-designed three-step process for learning a mapping between text representations and preference vectors.

**Weaknesses:**

1. **Limited novelty.** The primary novelty lies in the training procedure used to map sentence embeddings to preference vectors. While clearly presented, the three-step process mainly involves dataset construction and mean-squared-error regression, without introducing new algorithmic components or tackling novel challenges.
2. If trained from scratch, this method requires training 3 different models disjointly (3-stage training).
3. **Potential collapse.** The current procedure does not explicitly prevent the learned mappings from collapsing nor truly aligns the texts with the desired characteristics, just maps (see Question 2).

**Questions:**

1. In Line 187, it is mentioned that a direct end-to-end training would involve an intractable joint optimization problem. Why is this the case? Would it not be feasible to train the hypernetwork using sentence representations (or their projections) as preference vectors, while enforcing alignment between the generated output and desired characteristics?
2. Step 1.4 of Section 3.4 selects, for each scenario, the set of preference vectors that yield the best performance.
   1. Given this heuristic, a single vector could belong to the optimal set of multiple scenarios. Have you verified this empirically? Figure 2 shows considerable overlap between some clusters (e.g., light orange and purple, blue and green).
   2. How do you ensure that this procedure yields disjoint and well-separated clusters?
   3. How do you guarantee that the mapping does not collapse, i.e., that all scenarios do not end up sharing the same set of vectors?
3. Line 358 states that the semantic mapping is “well structured.” However, the corresponding figure alone is insufficient evidence for this claim and should be supported by quantitative analysis.
   1. You could compute intra-cluster and inter-cluster distances. For each distribution type, compute the average distance between preference vectors corresponding to texts of the same type, and compare it to the average distance to vectors of different types.
   2. Another possible analysis is to test linear separability by labeling data by distribution type and performing KNN classification with k-fold cross-validation.
4. What are the training details for the hypernetwork? Was a pretrained hypernetwork used, or was it trained from scratch? If pretrained, what dataset and procedure were followed?
5. Line 343 mentions training with a KL divergence loss, whereas Equation (9) specifies an MSE loss. Which one is actually used?

**Minor comments that do not affect rating**
1. In Figure 3b, between which two distributions is the KL divergence computed?
2. In Figure 2, what do the connecting lines between the left and right sides represent?
3. Table 1 and Figure 3a appear to display the same information. What is the rationale for including both? Additionally, Line 354 should reference Figure 3a as well as Table 1.
4. Line 469 refers to Figure 3b but appears to mean only a portion of it—please clarify the intended reference.

---

### Note · Authors · 2025-11-14

I have read and agree with the venue's withdrawal policy on behalf of myself and my co-authors.